# Graph Random Forest: A Graph Embedded Algorithm for Identifying Highly Connected Important Features

**DOI:** 10.3390/biom13071153

**Published:** 2023-07-20

**Authors:** Leqi Tian, Wenbin Wu, Tianwei Yu

**Affiliations:** 1School of Data Science, The Chinese University of Hong Kong, Shenzhen 518172, China; leqitian@link.cuhk.edu.cn (L.T.); wenbinwu@link.cuhk.edu.cn (W.W.); 2Shenzhen Research Institute of Big Data, Shenzhen 518172, China; 3Guangdong Provincial Key Laboratory of Big Data Computing, Shenzhen 518172, China

**Keywords:** feature selection, random forest, gene network

## Abstract

Random Forest (RF) is a widely used machine learning method with good performance on classification and regression tasks. It works well under low sample size situations, which benefits applications in the field of biology. For example, gene expression data often involve much larger numbers of features (p) compared to the size of samples (n). Though the predictive accuracy using RF is often high, there are some problems when selecting important genes using RF. The important genes selected by RF are usually scattered on the gene network, which conflicts with the biological assumption of functional consistency between effective features. To improve feature selection by incorporating external topological information between genes, we propose the Graph Random Forest (GRF) for identifying highly connected important features by involving the known biological network when constructing the forest. The algorithm can identify effective features that form highly connected sub-graphs and achieve equivalent classification accuracy to RF. To evaluate the capability of our proposed method, we conducted simulation experiments and applied the method to two real datasets—non-small cell lung cancer RNA-seq data from The Cancer Genome Atlas, and human embryonic stem cell RNA-seq dataset (GSE93593). The resulting high classification accuracy, connectivity of selected sub-graphs, and interpretable feature selection results suggest the method is a helpful addition to graph-based classification models and feature selection procedures.

## 1. Introduction

With the widespread use of high-throughput technologies, more gene expression datasets are available for studying clinical outcomes in diseases. Accurate classification models based on gene expression data are essential for identifying disease mechanisms and potentially designing specific treatments and medical plans. Identifying genes that determine different subtypes is essential for understanding the underlying biological mechanisms of disease development and discovering new drug targets [1]. Therefore, developing new methods to predict disease outcomes and identifying important features that reveal biological mechanisms is of great interest. However, the challenge of analyzing gene expression datasets lies in their structure, which often contains tens of thousands of genes and only a few hundred observations.

Tree-based methods such as random forest and LightGBM provide a natural solution to this problem, requiring little parameter tuning or data transformation [2]. Random forest, in particular, provides an inherent measure of feature importance, making it a popular choice for feature selection in the biological field [3]. Random forest algorithms have been shown to achieve high classification accuracy in distinguishing benign breast cancer from malignant [4], detecting biomarkers for prostate cancer progression based on DNA methylation data [5], and identifying abnormal pap smear cervical cell images [6]. Random forest is also commonly used in predicting gene regulatory networks (GRNs). For instance, the GENIE3 algorithm applies random forest to infer GRNs by solving a regression model based on target genes and selecting the strongest predictor as the regulator [7].

Gene expression data can benefit from incorporating network structure information, as functionally related genes tend to be dependent and close on the gene interaction network. Networks such as gene interaction networks, metabolite networks, miRNA networks, and protein–protein interaction (PPI) networks provide valuable information for disease prediction and can improve predictive performance [8]. More importantly, using networks to guide feature selection can result in models that are more interpretable. Integrating gene expression data with PPI networks can identify a more accurate subnetwork with markers [9]. Random forests have successfully incorporated network structure information in various ways. For example, IRatNet [10] utilizes heterogeneous data, including the PPI network, to derive preliminary information and integrate the information into a weighted sampling scheme under random forest to infer the final gene regulatory network. In the task of identifying predictive disease-related long non-coding RNA, GAERF [11] first embeds graph information with observed expression data into low dimensions using Graph Auto Encoder (GAE) and then performs random forest to predict the outcome. Neural networks can also be utilized to incorporate network structure information, such as GEDFN [12], which embeds a gene interaction graph in a feed-forward neural network, and GLRP [13], which uses a graph convolutional neural network to learn gene expression data with graph structure and then selects important features through the layer-wise relevance propagation (LPR) method. However, due to the small sample size, the over-parameterized problem in biological data remains a challenge for neural networks and requires more effort during training.

Most existing random forest methods integrate graph information in prior knowledge and do not alter the tree-building procedure. Additionally, the topological properties of the sub-graph created using selected features are often overlooked. Barabási [14] proposed a model for disease prediction using a network-based approach and demonstrated that disease-related components tend to be in proximity to already identified ones. In a protein–protein interaction network, functionally related genes are closer or directly connected to each other [15]. Ideally, a feature selection method should select features that form cliques on the entire graph. However, important features chosen by a random forest models are scattered on the feature network, which is inconsistent with our expectations. To address this issue, we propose a Graph Random Forest (GRF) model that includes network information in the tree-building process and identifies clustered important features on the input network. Similar to our work, network-guided forest (NGF) [16] uses graph information to construct a tree. However, NGF has limitations. Firstly, limiting the splitting scope in the neighborhood reduces flexibility as the neighborhood size for each gene is only sometimes significant. Secondly, the method requires prior knowledge to determine the range of head-splitting nodes, while randomly selecting the splitting node can increase the randomness of model performance. Thirdly, the algorithm is time-consuming as it needs to determine the scope of available splitting nodes each time. Fourthly, the graph structure selected by important features needs to be analyzed. Therefore, our approach focuses on improving the flexibility and robustness of random forests while examining the connectivity of feature selection sub-graphs.

## 2. Materials and Methods

### 2.1. Graph Random Forest

Our proposed method incorporates graph information in the process of generating each decision tree in a random forest. This approach is based on two key assumptions: first, that only a small fraction of features that form a sub-graph effectively affect the outcome, and second, that features in a graph are dependent on their neighborhoods. These assumptions have been concluded from real data and are widely used in previous works reviewed in Section 1.

We have developed a new method called Graph Random Forest (GRF) to effectively integrate graph information into the random forest framework. The architecture of the model is shown in Figure 1. The key idea behind GRF is to embed graph information in the process of building a decision tree. When establishing a decision tree, we consider features in the neighborhood of any number of hops to the head-splitting node. The head node for each decision tree is determined through a data-driven approach, allowing for a flexible and robust model that leverages the underlying graph structure. The source code is available as a public repository on Github (https://github.com/tianlq-prog/GRF (accessed on 26 June 2023)).

### 2.2. Evaluation of Feature Importance

In addition to predicting labels on testing data, it is crucial to identify features that significantly contribute to the classification and help uncover biological mechanisms. To this end, we leverage the Gini importance metric used in the random forest and adapt it for GRF.

In our implementation, we construct a random forest with $c_{i}$ decisions using features in the vicinity of each node i∈V, making it easy to compute feature importance by aggregating results from all forests. To compute feature importance, we initialize a vector $I={\left[{\mathrm{imp}}_{1},{\mathrm{imp}}_{2},\cdots ,\mathrm{imp}p\right]}^{T}$ with zeros for each element. For each node *i* with non-zero $c_{i}$, we build a random forest $F_{i}$ with $c_{i}$ trees using nodes in Neighbor(v,k). From $F_{i}$, we obtain the Gini importance $I_{i}=\left[{\mathrm{imp}}_{j}^{\left(i\right)}\right],\mathrm{for}j\in \mathrm{Neighbor}\left(v,k\right)$, where ${\mathrm{imp}}_{j}^{\left(i\right)}$ represents the Gini importance of node *j* estimated from forest $F_{i}$. We update *I* as ${\mathrm{imp}}_{j}=\mathrm{imp}j+{\mathrm{imp}}_{j}^{\left(i\right)}\times c_{i}$ for j∈Neighbor(v,k). This process allows us to access the feature importance for all variables, enabling us to select high-ranking features.

### 2.3. Details of Model Setting

The training of the GRF algorithm consists of two parts. In the first part, we trained a simple random forest where each tree had a depth of one. Using this random forest fitted on the training dataset, we recorded the number of occurrences of each node as a head-splitting node and embedded this information in GRF. Then we trained GRF on the training dataset with 500 trees and default parameter values in the second part. We did not fine-tune for the best model on specific data as our primary interest was in evaluating the ability to find important variables and studying the clustering properties of the selected sub-graphs.

### 2.4. Simulation Setting

To simulate disease classification, we conducted a series of experiments using gene expression data and a gene network. We compared the performance of GRF with that of the standard random forest. The simulation study aimed to examine whether GRF could identify an effective sub-graph more accurately.

#### Synthetic Data Generation

In our simulation studies, we generated a scale-free graph for *p* features using the Barab’asi–Albert (BA) model [17], which captures the degree distribution of biological networks characterized by a power law with parameter *m*. We set the power law parameter to 0.5 in our simulations. We then used the generated feature network to calculate the shortest distances between all pairs of vertices, which we represented a matrix $D\in R^{p\times p}$. Finally, we derived the covariance matrix Σ for the *p* features using the formula:(1)$$\Sigma_{i,j}=0.8^{D_{i,j}},i,j=1,\cdots ,p.$$

The diagonal elements of Σ were one since the distance from one node to itself was zero. With the feature graph and covariance matrix Σ, we simulated expression data $X={\left[x_{1},x_{2},\cdots ,x_{n}\right]}^{T}$ with *p* features. A multivariate Gaussian distribution with mean zero was used to generate *X*, for each sample $x_{i}$,
(2)$$x_{i}∼N\left(0,\Sigma \right),i=1,2,\cdots ,n.$$

In this way, two features tended to have more significant covariance if they were close to each other.

We adopted a strategy that favored significant covariance between two features if they were close to each other in the generated network. This approach allowed us to select a subset of true predictors that generated the outcome *Y*, corresponding to different disease outcomes. To identify potential core features, we first ranked the features according to their node degree, denoting it as $d_{1}\geq d_{2}\geq ,\cdots ,\geq n_{p}$. We then used an average strategy to calculate the average degree in three steps as ${\bar{d}}_{i}=\left(d_{i}+d_{i-1}+d_{i-2}\right)/3,i=3,4,\cdots ,p$. We determined the change point as the first feature whose degree $d_{t}$ decreased more than 10% compared to the averaged value, which was $\left(d_{t}-d_{t-1}\right)/{\bar{d}}_{i}>-0.01$. Finally, we chose potential core features with a node degree more significant than $d_{t}$. Our approach considered the scales and characteristics of different graphs, which might vary, and prioritized the identification of features with significantly higher node degrees.

In the simulation study, we conducted experiments with randomly chosen core features from the pool of potential core features. To expand the clique starting from the core features, we iteratively chose at most *m* neighboring features each time. To avoid the selected sub-graph becoming too dense, we assigned an attenuation rate to the parameter *m*. This process produced a collection of true predictors *S*, which formed a sub-graph. We denoted the size of *S* as $p_{0}$, and we sampled the parameter $\beta ={\left(\beta_{1},\beta_{2},\cdots ,\beta_{p_{0}}\right)}^{T}$ from a uniform distribution ranging from 0.1 to 0.8, with some of the values set to negative. We generated the output *Y* using a generalized linear model as follows:(3)$$P\left(y_{i}=1|x_{i}\right)=\sigma \left(x_{i}^{T}\beta +\beta_{0}\right),i=1,2,\cdots ,n$$
(4)$$y_{i}=I\left(P\left(y_{i}=1|x_{i}\right)>0.5\right),i=1,2,\cdots ,n$$
where σ was the logistic link function or absolute link function,
(5)$$\sigma \left(x\right)=\frac{1}{e^{x}+1}\;\mathrm{or}\;\sigma \left(x\right)=\mathrm{abs}\left(x-\bar{x}\right)+0.5.$$

To evaluate the performance of our proposed method GRF, we conducted a comparative study with RF using expression data *X* consisting of 4000 features and 500 samples. In real gene data, the true predictors are typically a small fraction of the total number of genes, typically around 0.5%. To simulate this scenario, we generated *Y* with varying numbers of true predictors, namely 30, 60, 90, 120, 150, 180, and 210. In addition to the number of true predictors, we also considered different sub-graph shapes. While a clustered sub-graph is typically formed by one clique, in some cases, it may appear as two cliques with minimal connection between them. Therefore, we evaluated the performance of our method on two different clustering shapes of the sub-graph, namely, one core and two cores. In the case of two core features, we ensured that the distance between the two core features was no less than four to obtain two distinct cliques. We used logistic and absolute link functions to test the GRF method’s effectiveness under different relationship assumptions between *X* and *Y*.

### 2.5. Real Datasets

For our study, we obtained gene expression data from two different datasets: the Cancer Genome Atlas (TCGA) and the Gene Expression Omnibus (GEO). Specifically, we used data from the TCGA for two common subtypes of lung cancer—lung adenocarcinoma (LUAD) and lung squamous cell carcinoma (LUSC). These datasets were accessed through the TCGA website (https://www.cancer.gov/ccg/research/genome-sequencing/tcga (accessed on 26 June 2023)). In addition, we utilized the human embryonic stem cell RNA-seq dataset (GSE93593) from the GEO database (https://www.ncbi.nlm.nih.gov/geo/query/acc.cgi?acc=GSE93593 (accessed on 26 June 2023)).

## 3. Results

### 3.1. Simulation Results

In our simulation study, we first trained a simple random forest with one-depth trees to determine head nodes in GRF and then built up 500 graph-embedded decision trees. For the vanilla RF model, we also set the number of trees to 500, with a maximum depth of 10 for each tree.

In our simulation, we conducted experiments using Python as our programming language. For each simulation setting, we generated ten datasets and then applied GRF and RF. Each dataset was randomly divided into a training set and a testing set using a 7:3 ratio. The model was trained on the training set and made predictions on the testing set. For each experiment, the computation time for GRF was around 7 s on a Linux workstation with a 5950× CPU, 128 Gb RAM, and a GTX 3060 GPU. In the simulation study, we were access to the specific list of true predictors contributing to the output *Y*. To evaluate the classification models, we employed test accuracy as the performance metric. This measure quantifies the classification ability of the models by comparing their predictions against the ground truth labels. To estimate the power of estimated feature importance, we utilized two widely recognized metrics: Area Under the Curve (AUC) and Precision-Recall AUC (PR-AUC). These metrics were calculated based on the estimated feature importance values and the ground truth effective feature list. The AUC provides a comprehensive measure of overall model performance, while PR-AUC focuses on the relationship between precision and recall. By incorporating both metrics, we aimed to obtain a comprehensive assessment of feature importance.

Figure 2 shows the results with logistic link function. The first row demonstrates the performance with a true sub-graph extended from one core node, and the second row shows the results of a true sub-graph extended from two core nodes. The error bar represents the estimated standard deviation estimated from ten experiments. Figure 2a,d show the test accuracy using RF and GRF with hopping steps 1, 2, and 3. In the case of one core node, GRF with hopping step 3 had the highest test accuracy when the number of true predictors was less than 100, and GRF 2 was higher when the size of true predictors grew larger. RF achieved the highest score in the setting with 210 true predictors. In the case of two core nodes, GRF with hopping steps 2 and 3 achieved the highest score in more than half of the cases. In general, when the hopping step was set to one, the test accuracy was lower compared to random forest. However, when the hopping step was set to two, both methods showed similar performance. Notably, when the hopping step was increased to three, our method outperformed random forest in terms of performance. Overall, there is minimal variation in test accuracy among the different methods, and an upward trend in prediction accuracy is evident as the number of true predictors increases.

The second column and third column of Figure 2 are AUC and PR-AUC with different sizes of core nodes. The estimated feature importance and a list with true predictors as one and other features as 0 were used for calculating the two metrics. The results of AUC using two/three hopping step GRF in one core simulation were mostly higher than 0.9, which was a huge improvement compared to the value in RF, which is around 0.6–0.75. Simulation results using the absolute link function are shown in Figure 3, which shows similar patterns. In general, we observed that a higher hopping step resulted in better feature selection performance. Overall, the simulation results verify that GRF has the capability to identify more significant features without sacrificing prediction accuracy.

To explore the property of the sub-graph, which consists of high-ranking features, we selected the top 100 most important features and extracted their connections from the simulated network. Figure 4 shows the sub-graph density, number of connected components, and size of the largest connected component in the setting with the logistic link function. The two rows correspond to settings of the true sub-graph extended from one or two core nodes. Figure 4a,d show the graph density, which is defined as the ratio between the number of edges and the number of all possible edges. A larger density indicates a more connected graph. Results show that GRF has a larger density in all experiments. Figure 4b,e present the number of connected component. A connected component means each pair of nodes in the component are connected through a certain pathway. A graph with more connected components indicates it is more scattered. Figure 4c,f show the size of largest connected component in different experiments. It is clear from the results that sub-graphs selected by RF are more separate, and the largest connected components are smaller. On the contrary, sub-graphs generated using GRF are highly connected and have large connected components. When the hopping step equals two or three, the largest connected component contains more than 80 nodes in different experimental settings. Results of using the absolute link function are shown in Figure 5, which demonstrate similar patterns.

Concerning the robustness in reproducibility of feature selection using GRF, we were curious whether selected features were stable across different model training times. To explore this property, we simulated a dataset with 500 samples, 4000 features, and 210 true features that determined the classes *Y* using a logistic link function. A GRF with hopping step two was used here. We repeated the training 20 times and recorded the top 100 features each time. In the 20 sets of most important features, 58 appeared more than 14 times, and 86 appeared more than ten times. Among the 58 features that appeared more than 14 times, 53 were on the list of true predictors, and the other five features were within the neighborhood of one step to true predictors. Among the 86 features that appeared more than ten times, 75 were true predictors, and the others were all around the true predictors in one step. For the union of the 20 lists of selected features, 46.4% of them were true predictors, and 89.3% of them were within one step neighborhood of true predictors.

### 3.2. Non-Small Cell Lung Cancer Data Results

We applied our GRF method to distinguish two types of most common subtypes of lung cancer—lung adenocarcinoma (LUAD) [18] and lung squamous cell carcinoma (LUSC) [19] from The Cancer Genome Atlas. Lung cancer is one of the deadliest cancers nowadays. However, it is still unclear how these two subtypes differ in biological mechanisms, and they are still treated equally as non-small cell lung cancer (NSCLC). We tried to identify the differences between these two subtypes and analyze the biological mechanisms using GRF. The LUAD dataset consisted of an expression matrix with 23,032 miRNA expressions in 524 patients, and the LUSC dataset contained an expression matrix with 23,652 miRNA expressions in 496 observations. Combining the two datasets of different subtypes using overlapped features and selecting the largest connected component of miRNA network obtained from HINT [15], we eventually obtained an expression matrix with 9819 miRNA features of 1020 samples. The label for LUAD was marked as one and LUSC as zero, correspondingly. We used the processed data for downstream analysis.

Using the data with their corresponding network, we conducted GRF and RF for comparison. When training the model, we split the dataset into the training set and testing set with the ratio of 7:3. We used GRF with a hopping step of two and evaluated the performance based on an averaged score. For each method, we conducted twenty experiments and summarized the performances. The first row in Table 1 shows that the two methods had good performance on the classification task with high predicting accuracy. The accuracy of the testing dataset for GRF and RF was around 0.94, indicating a different expression pattern of miRNA. It was reasonable to distinguish the two subtypes of lung cancer using this dataset. From the result, we figured out that further accuracy improvement is difficult and meaningless since the performance almost reached an upper bound. Though our proposed GRF had a tiny gap in accuracy compared to RF, GRF had advantages in identifying potentially important features which had been proved in the simulation study. Moreover, features selected by GRF were more likely to fall into cliques on the original graph and could generate a sub-graph with fewer connected components. A highly connected graph was preferred because of its similarity to the ones found in experimental studies. By choosing 100 features with the highest importance scores, we generated the sub-graph shown in Figure 6a. Table 1 also exhibits selected sub-graph properties through GRF and RF. The second row is the number of connected components, and the third row represents the size of the largest connected component. The fourth and fifth rows are the average shortest distance and the average distance in the largest components between each node. The results illustrate that the sub-graph selected by GRF had fewer connected components and possessed a larger connected component with a reasonable distance.

Functional analysis of genes selected by GRF was conducted by testing the enrichment of gene ontology (GO) biological processes using the clusterProfiler package [20]. The biological processes with *p*-values less than 0.01 and adjusted *p*-values less than 0.05 were considered significant pathways in our study. The top 15 GO terms are shown in Table 2. Twelve of the 100 selected genes belonged to the regulation of DNA metabolic process, which was the most important GO term. In addition, ‘DNA ligation’, ‘somatic DNA recombination’, and ‘regulation of DNA biosynthesis processes’ were among the top terms. Changes in DNA function and damage affect cell proliferation and differentiation, which may influence cancer progression [21]. Meanwhile, overexpression of CDK2 and CDK16 in LUSC has been proved to cause abnormal regulation of cell cycle and promote cell proliferation. These effects may increase the malignant potential of the tumor and lead to a faster growth speed compared to the progression of LUAD [22,23]. The second-most important term was ‘telomere maintenance’, and there were many terms among the top ones related to telomeres, such as ‘telomere organization’, ‘negative regulation of telomere maintenance via telomerase’, and ‘telomere elongation control’. Limiting telomeres from shortening is one of the significant mechanisms by which cancer cells gain resistance to inhibition. The ability to maintain telomeres above a critical length represents the degree of cell deterioration [24]. Cancer cells can resist death and realize the immortality of replication through activating the telomerase [25]. LUAD and LUSC have different expression levels of telomere-related genes, so telomere maintenance genes are considered to be potential biomarkers for two subtypes. At the same time, a vaccine against telomerase named GV1001 has been proved to be beneficial to immunotherapy for NSCLC patients [26].

The remaining significant GO terms include ‘T cell proliferation and activation’, ‘immune response modulation’, ‘embryonic morphological development’, ‘endoderm development’, and so on. Studies on immune-related genes (IRGs) have found that T-cell receptor signaling expresses differently in two subtypes [27]. The MHC molecule, which is crucial for antigen processing in immune responses, and chemokine, which guides cell migration, are found to be inhibited more rapidly in LUSC. These observations confirm that LUSC grows faster by suppressing the immune system. HOX gene encoding is an important transcription factor in the embryonic development and differentiation of adult cells. Recent studies have shown that HOXA1 is significantly up-regulated and hypermethylated in LUAD [28]. The genes Hh and ErbB are found to be strongly correlated with two subtypes, which are related to lung development [29]. Hh maintains stem cells, responds to injury, and affects the formation of bronchial numbers. ErbB can cause defects of type II epithelial cells in the alveolar lining and reduce branching morphogenesis in embryos by affecting the expression level of anti-EGF antisense oligonucleotides. The whole table containing all GO terms for functional analysis using GRF and RF can be found in https://github.com/tianlq-prog/GRF/blob/main/Supplementary.pdf (accessed on 26 June 2023).

### 3.3. Human Embryonic Stem Cell Data Results

We also applied GRF on the human embryonic stem cell RNA-seq dataset from GEO (GSE93593) [30]. The dataset contained 23,045 genes from 1733 observations and their corresponding clinic information, including doublecortin (DCX) status and days of culture. We obtained the gene network from the HINT database [15]. After screening the genes in the HINT database and selecting the largest connected component of the network, 12,215 genes were finally selected. A log transformation was conducted on the expression value of each gene. Our goal was to explore the relationship between gene expression and the status of DCX, whether positive or negative. Therefore, the task became a binary classification problem.

Doublecortin (DCX) is a microtubule-associated protein expressed explicitly by immature neurons in embryonic and adult cortical structures. It is necessary for neuron migration and differentiation and is closely related to the development of the central nervous system. Since the expression of DCX changes two weeks before the appearance of new neurons, the richness of neurogenesis in the brain cannot be directly quantified. So, DCX is a powerful tool for identifying early and immature neurons. Therefore, research on the transient expression of DCX to help understand the development of the nervous system has received extensive attention.

We tested GRF and RF on a human embryonic stem cell dataset with twenty repeated experiments. The computation time for each experiment using GRF on the workstation was around 100 s. The classification accuracy results are shown in the first row of Table 3. The mean accuracy was high for each method, and GRF had slightly lower accuracy than RF. However, as shown in Table 3, the sub-graphs generated by GRF with the top 100 most important genes had higher connected properties than the ones using RF. Specifically, when using GRF, the number of connected components was much smaller, and the sub-graphs had larger connected components with an average value achieving 67.

Figure 6b is the sub-graph of the top 100 genes with the highest averaged importance score using GRF. GO enrichment analysis was performed on the sub-graph, and the top 10 pathways are shown in Table 4. The top GO term was ‘positive regulation of epithelial to mesenchymal transition’. In addition, ‘homotypic cell–cell adhesion’ and ‘epithelial cell differentiation’ were among the top terms. Epithelial cells have regular cell–cell contacts and adhesion to surrounding cellular structures, thus can avoid the separation of individual cells [31]. However, during embryonic development, cells need to migrate to adjacent tissues to form new organs, and tissues [32], so quiescent epithelial cells undergo epithelial–mesenchymal transition (EMT), thereby differentiate into motile mesenchymal cells [33] and possess the invasive ability. The response to EMT comes from stromal cells such as fibroblasts and mesenchymal stem cells. These stromal cells secrete a series of heterotypic signals, growth factors, platelet-derived growth factor (PDGF), and epidermal growth factor (EGF) [34]. This also explained the appearance of ‘platelet aggregation’, ‘regulation of hematopoiesis’, and ‘platelet activation’ in the top GO terms. Also, many of the genes involved in ‘lung development,’ ‘respiratory system development,’ and ‘air duct development’ are part of the response to growth factor stimulation, leading to the significance of these terms. The second most significant GO term was ‘synaptic organization’, and other important terms related to it included ‘axon development’ and ‘axogenesis’. Synapses, responsible for transmitting information between neurons and target cells, play an essential role in nervous system development. The fetal brain begins to develop from the third week of gestation [35], neural precursor cells divide and form neurons and glia. Furthermore, the number of synapses keeps increasing in the first few years of life [36]. The fifth-ranked GO term was ‘transmembrane receptor protein serine/threonine kinase signaling pathway’. The serine–threonine kinase Akt plays a central role in integrating cellular responses to growth factors [37], and it has been proven to maintain cellular integrity and protect ‘tagged’ from exposure. It is also involved in the phagocytic disposition of cells, in which it promotes neuronal and vascular survival and prevents induction of programmed cell death [38]. Overall, in the DCX classification task, GRF could identify important and easily interpretable sub-graphs.

## 4. Discussion

In this study, we proposed a novel method, Graph Random Forest (GRF), which integrates graph information into the random forest framework to improve the accuracy and interpretability of the model. We applied GRF to simulation data and two real datasets: one involving the classification of two subtypes of lung cancer and the other involving the classification of doublecortin status in human embryonic stem cells. Our results demonstrated that GRF had comparable classification accuracy to the traditional random forest method while generating a sub-graph with higher connectedness and providing more interpretable feature importance scores.

Our proposed method leverages the underlying graph structure of the data, which captures the dependencies and interactions between the features. By incorporating the graph information into the decision tree construction process, GRF can effectively identify and utilize the most informative features while reducing noise and irrelevant features. Our results showed that GRF was able to identify potentially important features that were more likely to fall into cliques on the original graph and could generate a sub-graph with fewer connected components, which are more similar to those found in experimental studies.

The biological insights obtained from our experiments provide evidence for the effectiveness of GRF in identifying important features and their interactions. For example, in the lung cancer dataset, GRF identified genes related to DNA metabolic processes and telomere maintenance, which are known to be associated with cancer progression and resistance to inhibition. In the human embryonic stem cell dataset, GRF identified genes related to epithelial-mesenchymal transition, axon development, and synapse organization, which are critical for nervous system development. These findings demonstrate the ability of GRF to uncover important biological mechanisms and potential biomarkers that could be useful for further experimental validation.

In summary, our proposed method, GRF, provides a powerful tool for analyzing complex data with graph structure. By incorporating graph information into the random forest framework, GRF can effectively improve the accuracy and interpretability of the model, as demonstrated by our experiments on lung cancer and human embryonic stem cell datasets. We believe that our method can also be widely applied to other fields, such as social network analysis and image analysis, where the data have an inherent graph structure.

## 5. Conclusions

We presented a new version of random forest which embeds graph information. It has the ability to identify important features with the property of high connectivity in the original information graph. A simulation study verified that our method could find features with higher AUC and PR-AUC without losing much classification accuracy. Two real applications showed the power of selection, which helps reveal meaningful biological mechanisms.

## Figures and Tables

**Figure 1 biomolecules-13-01153-f001:**
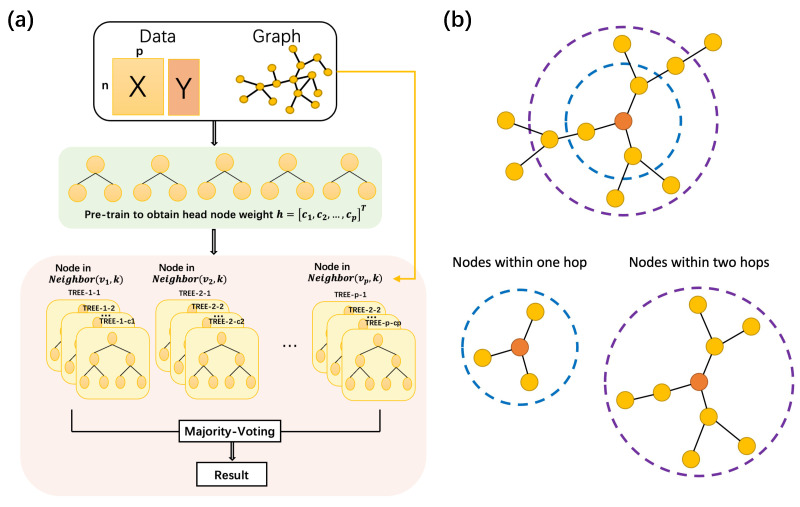
(**a**) The overall architecture of Graph Random forest; (**b**) Visualization of one-step hop vicinity and two-step hops vicinity.

**Figure 2 biomolecules-13-01153-f002:**
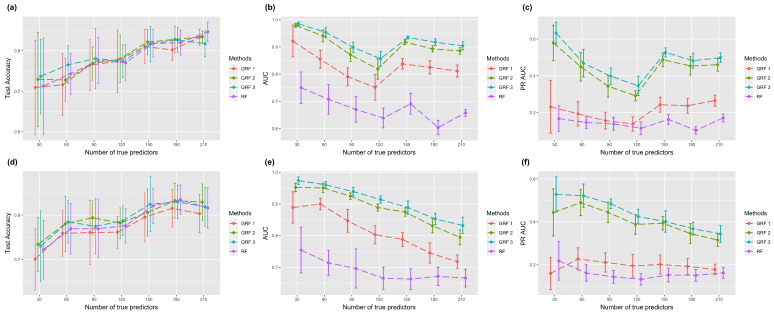
Plots of classification and feature selection with the logistic link function. The first row corresponds to one core node, and the second corresponds to two core nodes. Error bars represent the mean value plus/minus the standard error. GRF with different numbers indicates various parameters of selective range. (**a**,**d**) Test accuracy of classification task. (**b**,**e**) AUC of feature selection. (**c**,**f**) PR-AUC of feature selection.

**Figure 3 biomolecules-13-01153-f003:**
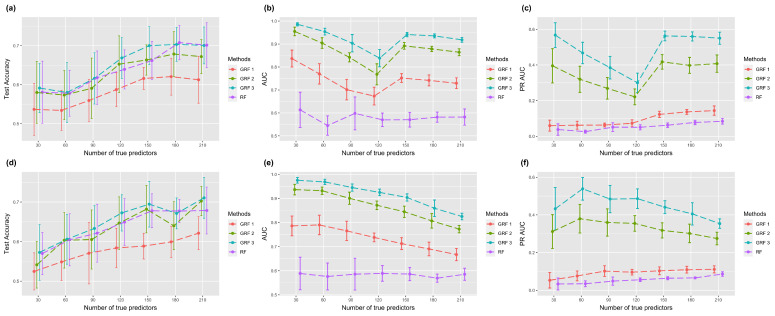
Plots of classification and feature selection with the absolute link function. The first row corresponds to one core node, and the second corresponds to two core nodes. Error bars represent the mean value plus/minus the standard error. GRF with different numbers indicates various parameters of selective range. (**a**,**d**) Test accuracy of classification task. (**b**,**e**) AUC of feature selection. (**c**,**f**) PR-AUC of feature selection.

**Figure 4 biomolecules-13-01153-f004:**
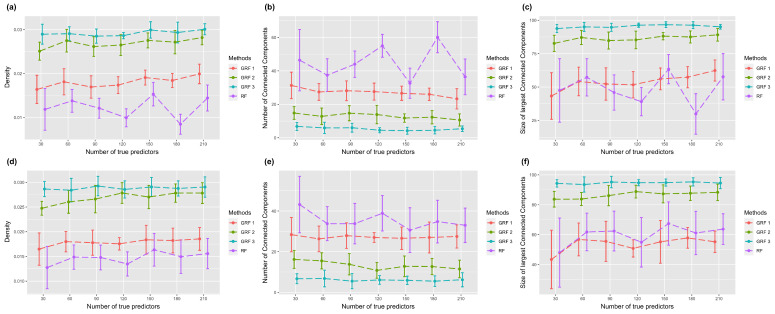
Plots of sub-graph properties with the logistic link function. The first row corresponds to one core node, and the second corresponds to two core nodes. Error bars represent the mean value plus/minus the standard error. GRF with different numbers indicates various parameters of selective range. (**a**,**d**) The density of selected sub-graph. (**b**,**e**) The number of connected components of selected sub-graph. (**c**,**f**) The size of connected components of selected sub-graph.

**Figure 5 biomolecules-13-01153-f005:**
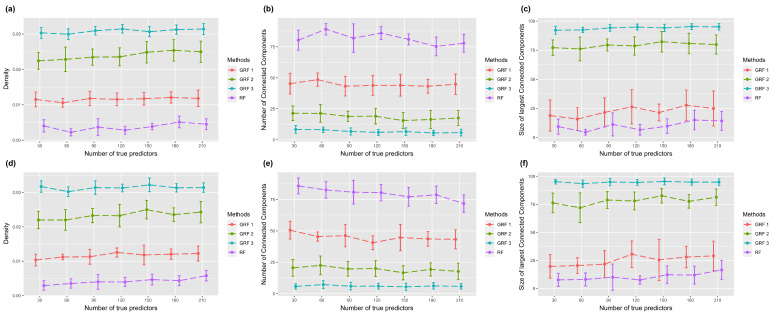
Plots of sub-graph properties with the absolute link function. The first row corresponds to one core node, and the second corresponds to two core nodes. Error bars represent the mean value plus/minus the standard error. GRF with different numbers indicates various parameters of selective range. (**a**,**d**) The density of selected sub-graph. (**b**,**e**) The number of connected components of selected sub-graph. (**c**,**f**) The size of connected components of selected sub-graph.

**Figure 6 biomolecules-13-01153-f006:**
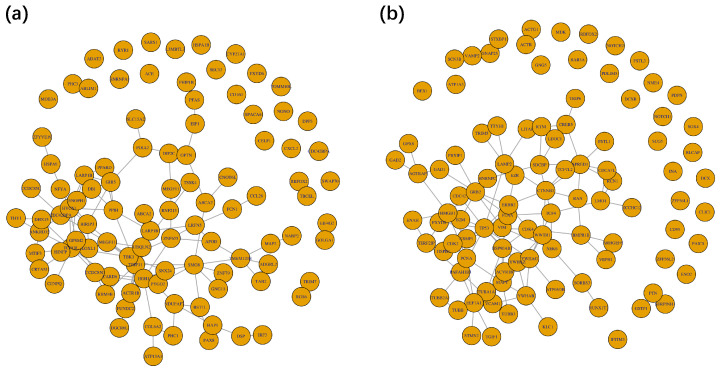
Sub-graph selected by GRF on real dataset. (**a**) On non-small cell lung cancer dataset; (**b**) On human embryonic stem cell dataset.

**Table 1 biomolecules-13-01153-t001:** Classification result and properties of selected sub-graph for NSCLC data.

Methods	GRF ^1^	RF ^1^
Mean accuracy	0.9457 (0.0116)	0.9483 (0.0097)
Number of connected components	20.65 (3.63)	94.9 (1.92)
Size of the largest connected component	73.75 (6.63)	3.7 (1.22)
Average distance	4.29 (0.25)	1.38 (0.38)
Average distance in the largest component	4.31 (0.25)	1.53 (0.33)

^1^ The values in brackets correspond to the standard deviations.

**Table 2 biomolecules-13-01153-t002:** Top 10 GO biological process for the sub-graph selected by GRF on NSCLC data.

GOBPID ^1^	Adj-*p* ^2^	Term
GO:0051052	0.015	regulation of DNA metabolic process
GO:0000723	0.015	telomere maintenance
GO:0032069	0.015	regulation of nuclease activity
GO:0032200	0.015	telomere organization
GO:0032211	0.015	negative regulation of telomere maintenance via telomerase
GO:0051098	0.015	regulation of binding
GO:0048598	0.028	embryonic morphogenesis
GO:1904357	0.028	negative regulation of telomere maintenance via telomere lengthening
GO:0042098	0.028	T cell proliferation
GO:0006303	0.028	double-strand break repair via nonhomologous end joining

^1^ Manual pruning of partially overlapping GO terms was performed. ^2^ Adj-*p* represents the adjusted *p*-value.

**Table 3 biomolecules-13-01153-t003:** Sub-graph property for GSE data. The values in brackets correspond to the standard deviations.

Methods	GRF ^1^	RF ^1^
Mean accuracy	0.9280 (0.0089)	0.9301 (0.008)
Number of connected component	31.15 (4.83)	83.85 (3.73)
Size of the largest connected component	67.00 (6.10)	7.95 (3.32)
Average distance	3.67 (0.30)	2.17 (0.62)
Average distance in the largest component	3.68 (0.30)	2.54 (0.62)

^1^ The values in brackets correspond to the standard deviations.

**Table 4 biomolecules-13-01153-t004:** Top 10 GO biological process for the sub-graph selected by GRF on DCX data.

GOBPID ^1^	Adj-*p* ^2^	Term
GO:0010718	0.0002	positive regulation of epithelial to mesenchymal transition
GO:0050808	0.0002	synapse organization
GO:0010717	0.0002	regulation of epithelial to mesenchymal transition
GO:0034109	0.0002	homotypic cell–cell adhesion
GO:0007178	0.0002	transmembrane receptor protein serine/threonine kinase signaling pathway
GO:0070527	0.0003	platelet aggregation
GO:0048667	0.0003	cell morphogenesis involved in neuron differentiation
GO:0048812	0.0003	neuron projection morphogenesis
GO:1903706	0.0003	regulation of hemopoiesis
GO:0001837	0.0003	epithelial to mesenchymal transition

^1^ Manual pruning of partially overlapping GO terms was performed. ^2^ Adj-*p* represents the adjusted *p*-value.

## Data Availability

The data are available from the public sources.

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
