# Peer review of "Graph Random Forest: A Graph Embedded Algorithm for Identifying Highly Connected Important Features"

_biomolecules, 2023, doi:10.3390/biom13071153_

Round 1

Reviewer 1 Report

While random forest has been widely applied in biology, RF lacks good feature selection ability. The genes selected in RF are loosely connected that conflicts with the biological assumption. In this work, the authors designed a new model, called Graph Random Forest (GRF), to improve feature selection with minimum accuracy loss. GRF can identify important features to form highly connected subgraphs. Through simulation experiments and two real datasets, GRF is proved to be a promising tool over RF in graph-based classification models and feature selection procedures. The manuscript is very well written and can be accepted in present form. 

Two minor points:

1. line 107, imp_ij, ij is not subscript.

2. I do encourage the authors to share their codes on GitHub to have a broader impact.

Reviewer 2 Report

The article  provides a detailed explanation of the simulation study and the evaluation of GRF and Random Forest (RF) methods. It includes information on the dataset generation, model training, computation time, evaluation metrics (ROC curve, AUC, PR-AUC), and the comparison of results between different methods. The subsection on the properties of the selected sub-graph and the functional analysis of genes selected by GRF adds depth to the analysis.The results show high classification accuracy, connectivity of selected sub-graphs, and interpretable feature selection results.

 Overall, the artice is well-structured and provides clear insights into the performance and capabilities of GRF.

1.     In the simulation study, what were the specific parameters considered for determining the selective range in GRF (hopping steps)? How did the choice of hopping steps affect the test accuracy and feature selection performance of GRF compared to RF?

2.     What were the parameters used in the simulation study to train the random forest and graph-embedded decision trees?

3.     ow many datasets were generated for each simulation setting, and how were they divided into training and testing sets?

4.     What hardware and computation time were used for the simulations? How were the power of estimated feature importance and the performance of classification models evaluated?

MINOR REVSION 
